# The Influence of Local Winds on Wind Power Characteristics in a High Arctic Valley

Matthias Henkies<sup>1,2</sup>, Knut Vilhelm Høyland<sup>2,1</sup>, Aleksey Shestov<sup>1</sup>, and Anna Sjöblom<sup>1</sup>

<sup>1</sup>The University Centre in Svalbard, 9171 Longyearbyen, Norway

<sup>2</sup>Norwegian University of Science and Technology, 7491 Trondheim, Norway

**Correspondence:** Matthias Henkies (matthiashe@unis.no)

Abstract. Wind power in the High Arctic is little developed, and understanding of the local wind conditions is needed. Therefore, the average wind characteristics in the Svalbard valley Adventdalen are investigated using primarily observations from wind profilers and weather stations. Low-level jets (LLJs) are frequent in calm synoptic conditions because the LLJs are, to a large degree, driven by local thermal gradients. Moreover, the LLJs increase the wind speed at low levels. The average observational wind profile has a wind speed maximum around the height of 80 m to 100 m and the wind power density (WPD) a maximum around 120 m to 140 m. This is poorly represented by numerical models and also differs from wind speed profiles from flat locations where the wind speed and WPD typically increase with height. The presence of valley winds is further responsible for a decreased speed variability at low levels, which leads to shortened periods with persistent little wind compared to higher elevated locations. The wind characteristics in Adventdalen advocate for the consideration of wind power in Arctic valleys.

## 1 Introduction

While the Arctic is heavily affected by global warming (e.g., Serreze and Barry, 2011; Previdi et al., 2021), this world region is also still highly dependent on fossil energy (de Witt et al., 2021), which again propels global warming. This is especially apparent in Svalbard (map in Fig. 1a), which is among the planet's fastest warming regions (e.g., Wickström et al., 2020), while most of its electricity has been produced by coal and currently diesel. In this setting Svalbard's largest settlement Longyearbyen transforms towards a renewable energy supply (Karlsen, 2025). Longyearbyen is located in the centre of Svalbard's largest island Spitsbergen, on the side of the large U-shaped valley Adventdalen (Figs. 1b, c).

As this location is at 78°N, it has a High Arctic climate with low temperatures and a pronounced yearly cycle. There are almost four months without sunlight in winter and an equally long period with continuous daylight in summer. Strong winds are more common in winter (Hanssen-Bauer et al., 2019). This makes wind power a potent possible energy source especially in winter, when no photovoltaic power can be produced. Research is nevertheless still needed to understand the underlying physical processes, how they differ from locations at lower latitudes and the effect they have on the wind power potential. Some of the reasons for these uncertainties are the scarce network of weather stations (e.g., Hanssen-Bauer et al.,

Figure 1. a: location of Svalbard in the Arctic; map data: Google, © 2024 Data SIO, NOAA, U.S. Navy, NGA, GEBCO, Landsat / Copernicus, IBCAO, U.S. Geological Survey. b: location of Adventdalen within Spitsbergen; map data: Google, © 2024 IBCAO, Landsat / Copernicus. c: 3D map of Adventdalen where circles mark weather stations. The local valley axis and wind directions at AD are illustrated with straight lines and arrows, respectively. Note that the view is from the north to display the plateau weather stations properly. Map data: TopoSvalbard © 2024 Norwegian Polar Institute

2019), boundary layer processes which are difficult to model (e.g., Mayer et al., 2012) and complex terrain, which is not fully resolved by current weather models (e.g., Valkonen et al., 2020).

The complex terrain of the Arctic makes the wind resources vary very locally (e.g. Schön et al., 2022). In general the wind speed on mountains is higher than in valleys because the wind is less obstructed at higher elevations and there is also a speed-up effect near ridges (e.g., Emeis, 2014a). However, mountains and ridges are also more prone to icing, i.e., the meteorological icing frequency increases with altitude and exposure to wind (Grünewald et al., 2012). Moreover, mountains are often less accessible, which is an obstacle for wind turbine construction. As these challenges can counteract the benefits of a greater wind resource on high mountains, valleys constitute an alternative location for wind power.

Although not in the Arctic, there are several examples of wind turbines which are placed in large alpine valleys in Switzerland, namely three in the Rhone valley and one in the Alpine Rhine valley (Hertach et al., 2024), which both are surrounded by peaks well over 2000 m above the valley floor. The turbines each have a capacity of 2 MW to 3 MW and produced energy with an average capacity factor of 0.17 (Rhine) and 0.26 to 0.29 (Rhone) from 2013 to 2024, which is comparatively high for Switzerland. This shows that some valleys can be suitable places for wind turbines.

One element of valley wind climates are thermally driven winds, i.e., anabatic and katabatic winds, which occur in synoptically calm conditions following a local pressure gradient based on a temperature gradient (e.g., Zardi and Whiteman, 2013). These winds usually develop with a diurnal rhythm and the radiative budget, vary in scale and strength and occur on slopes, along valleys and over mountain ranges. In Switzerland, the Rhone valley is especially known for relatively strong and frequent thermally driven valley winds (Schmid et al., 2020; Schmidli and Quimbayo-Duarte, 2023). The relevance of thermally driven winds for wind power has been investigated, e.g., in Switzerland (Kruyt et al., 2017) and at the North Sea coasts (Steele et al., 2015).

Dynamically-driven winds on the other hand are controlled through the pressure gradient and/or momentum at the large, synoptic scale (e.g., Whiteman and Doran, 1993). These winds include pure downward momentum transport, resulting in a valley wind aligning with the large-scale wind, as well as forced channelling, where the valley wind is driven by the large-scale momentum, but guided by the valley walls. Moreover, there is the pressure-driven channelling, during which the valley wind follows the synoptic pressure component along the valley axis. In addition to pure downward momentum transport, valley topography can lead to speed-up effects, such as gap winds, mountain waves and downslope windstorms (e.g., Jackson et al., 2013). Winds can be a combination of several of these mechanisms, even of both dynamic and thermal drivers as the Arctic Novaya Zemlya Bora shows (e.g., Efimov and Komarovskaya, 2018). This can sometimes make a clear attribution difficult.

Speed-up effects can create wind speed maxima at low heights, i.e., low-level jets (LLJs). However, there are several mechanisms which can lead to LLJs besides thermally or dynamically driven winds in complex terrain. LLJs are often attributed to inertial oscillations at the top of the nocturnal, stable boundary layer (e.g., Blackadar, 1957), or found in connection with fronts (e.g., Browning and Pardoe, 1973).

The properties of thermally and dynamically-driven winds have also been studied in Svalbard valleys. Henkies et al. (2023) found that a combined valley wind and sea breeze forms in Adventdalen (Fig. 1) in summer, while katabatic winds occur over glaciers and valleys in winter (e.g., Kilpeläinen et al., 2011; Esau and Repina, 2012). Low-level jets (LLJs) have been

observed in Adventdalen and connected to katabatic winds (Vihma et al., 2011). This, as well as the occurrence of very stable stratification, cold air pools and wind channelling in Adventdalen, have been confirmed through observations and simulations (Mayer et al., 2012; Valkonen et al., 2020; Mack et al., 2025). On a larger scale, Svalbard's mountain ranges, fjords and valleys can accelerate the large-scale flow locally on the lee side, as shown mostly for easterly wind directions (Skeie and Grønås, 2000; Sandvik and Furevik, 2002; Dörnbrack et al., 2010; Barstad and Adakudlu, 2011; Shestakova et al., 2022).

With many previous studies having explored wind phenomena, Adventdalen is a very suitable place to build on this knowledge and expand it towards wind energy applications. While the study of wind in Svalbard in general, and Adventdalen especially, has so far concentrated on case studies, this work shifts the focus to average multi-year wind conditions. With an emphasis on quantities relevant for wind power and using observational data from wind profilers and weather stations, the influence of local winds in Adventdalen on the wind climate is investigated. After the introduction of relevant quantities for wind power and wind profile extrapolation in Sect. 2, the methods are explained in Sect. 3. The results are presented and discussed in Sect. 4, starting with the occurrence of different wind conditions (regimes) and their connection to thermally driven winds, followed by the properties of low-level jets (LLJs) and their drivers and how they shape the wind speed and wind power density (WPD) profiles. Then the observed profiles are compared with numerical models, and finally the characteristics of weak-wind periods (WWPs) at different locations are shown. In Sect. 5 the findings are applied to practical considerations for wind power in complex Arctic terrain.

#### 75 2 Theory

Wind power density (WPD) is the kinetic power P of the wind crossing a unit area A at a given height z. WPD is very sensitive to the wind speed u, which varies with z:

$$WPD(z) = \frac{P(z)}{A} = 0.5\rho(z)u(z)^{3}$$
 (1)

where  $\rho$  is the air density. As measurements at the desired height are often not available, u(z) can be estimated through simple extrapolation laws. The easiest approach is the power law:

$$\frac{u(z)}{u(z_1)} = \left(\frac{z}{z_1}\right)^{\alpha} \tag{2}$$

where  $z_1$  is a reference height and  $\alpha$  is a fixed parameter, often chosen between 0.10 and 0.14. The logarithmic law (log law) has two parameters, the friction velocity  $u_*$  and the surface roughness  $z_0$ :

$$\frac{\kappa u(z)}{u_*} = \log(\frac{z}{z_0}) \tag{3}$$

with  $\kappa = 0.40$  the von Kármán constant. This extrapolation requires measurements at two heights to determine its parameters; if  $z_0$  is estimated, only one. Typical values of  $z_0$  for the Arctic tundra are of the orders of 1 mm for snow or 1 cm for grass and similar-sized vegetation (e.g. Stull, 1988). The log law is based on Monin-Obukhov Similarity Theory in the constant flux layer, whose depth is typically tens of metres, but highly variable. Moreover, the law must be modified to account for the

influence of stability (e.g., Stull, 1988). It has to be kept in mind that these two laws are based on measurements in flat terrain and may not be applicable in a valley like Adventdalen. Further, the height range over which the laws are valid is below the height of most potential wind turbine rotors.

### 3 Methods

The main focus is the middle of Adventdalen, AD (Fig. 1). There the valley has a ca.  $3 \,\mathrm{km}$  wide floor with steep sidewalls and a few small side valleys. Mountains surround the valley with peaks around  $800 \,\mathrm{m}$  to  $1000 \,\mathrm{m}$  altitude. South of Adventdalen there are also plateaus at an altitude of  $400 \,\mathrm{m}$  to  $600 \,\mathrm{m}$ .

### 3.1 Wind regimes

100

Wind regimes (WRs) are defined based on the combination of the observed  $10 \,\mathrm{m}$ ,  $1 \,\mathrm{h}$  average wind at AD and the large scale wind aloft (Table 1). To exclude any effects of the local topography in the large scale wind, the reanalysis ERA5 is used, since it has a coarse resolution of  $31 \,\mathrm{km}$ , which does not resolve a valley like Adventdalen. The ERA5 wind is evaluated at the closest gridpoint on the  $900 \,\mathrm{hPa}$  level. The wind direction ( $\theta$ ) is considered relative to the local valley axis of Adventdalen (Fig. 1c), which is from  $125^{\circ}$  (down-valley) or  $305^{\circ}$  (up-valley). This is performed over the  $10 \,\mathrm{a}$  period 2013-2022. The central idea behind this is to divide into local winds which have little connection to the large scale wind, i.e., *decoupled*, and local winds which are likely driven by the large scale wind, i.e., *aligned*. In addition, the local winds are divided into *up*(-valley) and *down*(-valley) direction, reflecting the general channelling of winds in Adventdalen, plus an *across*(-valley) category.

**Table 1.** Definition of WRs based on wind speeds (u) and directions  $(\theta)$ , and their frequency (f) over 10 a

| WR             | AD (valley) 10 m wind                                                                       | ERA5 $900\mathrm{hPa}$ wind at $78.25^{\circ}\mathrm{N},15.75^{\circ}\mathrm{E}$         |      |
|----------------|---------------------------------------------------------------------------------------------|------------------------------------------------------------------------------------------|------|
| Decoupled up   | $\theta$ within $305^{\circ} \pm 45^{\circ}$ (up-valley)                                    | $\theta$ within $125^{\circ} \pm 90^{\circ}$ (down-valley) or $u < 5\mathrm{ms^{-1}}$    | 0.14 |
| Decoupled down | $\theta$ within $125^{\circ} \pm 45^{\circ}$ (down-valley)                                  | $\theta$ within $305^{\circ} \pm 90^{\circ}$ (up-valley) or $u < 5\mathrm{ms^{-1}}$      | 0.32 |
| Across         | $\theta$ within $215^{\circ} \pm 45^{\circ}$ or $35^{\circ} \pm 45^{\circ}$ (across-valley) | any                                                                                      | 0.12 |
| Aligned up     | $\theta$ within $305^{\circ} \pm 45^{\circ}$ (up-valley)                                    | $\theta$ within $305^{\circ} \pm 90^{\circ}$ (up-valley) and $u \geq 5\mathrm{ms^{-1}}$  | 0.11 |
| Aligned down   | $\theta$ within $125^{\circ} \pm 45^{\circ}$ (down-valley)                                  | $\theta$ within $125^{\circ} \pm 90^{\circ}$ (down-valley) and $u \ge 5\mathrm{ms}^{-1}$ | 0.31 |

#### 105 3.2 Wind profiles

Wind vector profiles  $u(z)^{10 \text{ min}}$  with 10 min resolution were taken during two campaigns at AD:

- Sodar (Sound Detection and Ranging) measurements from October 2015 to September 2017, covering the altitude range between  $30\,\mathrm{m}$  and  $1000\,\mathrm{m}$  with  $10\,\mathrm{m}$  resolution (Henkies et al., 2024a). The Sodar data includes mostly low and medium wind speed cases with  $u(10\,\mathrm{m}) < 12\,\mathrm{m\,s^{-1}}$  because of the disturbing noise at higher wind speeds.

- Lidar (Light Detection and Ranging) measurements from February to June 2020 with 12 levels between 40 m and 290 m altitude (Henkies et al., 2024b). The Lidar was able to measure during high wind speed cases, while, with a CNR threshold of −22 dB, it occasionally had problems in bad visibility, independent of the wind speed.

The observations are processed and combined into 12 vertical levels with 1 h resolution. The original Lidar data with a  $10 \,\mathrm{min}$  resolution were first filtered. Intervals during which the Lidar was only partly running, e.g., due to a power cut, were excluded based on a flag in the  $10 \,\mathrm{min}$  data files. In addition only measurements with at least  $10 \,\%$  data availability were used. Then hourly vector averages were taken, but only kept if there were at least two (out of six) valid values.

The original Sodar data were not filtered. They were also vector-averaged over 1 h, with the vertical resolution adjusted to the Lidar's, namely 40, 60, 80, 100, 120, 140, 160, 180, 200, 230, 260, 290 m, by taking a weighted average of the same level and the neighbouring levels:

120 
$$u(z) = \frac{1}{4} (2u^{10\min}(z) + u^{10\min}(z - 10m) + u^{10\min}(z + 10m))$$
 (4)

The 1 h Sodar averages were kept if at least 50% of the weighted original data were available. The processing left between 11725 and 12985 Sodar and between 840 and 2355 Lidar datapoints at each level, with lowest coverage at the largest heights.

Finally the 12 levels of Lidar and Sodar were merged with the (vector-averaged)  $10 \,\mathrm{m}$  wind speed from the weather station at AD to form a set of wind speed profile observations over 13 levels, where  $u_v(z)$  denotes the absolute values of these vector-averaged wind speed profiles and u(z) the mean absolute speed, which is commonly used. This results in a combination of Sodar, Lidar and weather station observations which cover a a large variety of common wind speeds and directions in Adventdalen.

#### 3.3 Low-level jets (LLJs)

110

A low-level jet (LLJ) is here defined as a wind speed maximum  $u_{\rm LLJ}$  at height level  $z_{\rm LLJ}$ , with  $u_{\rm LLJ}$  at least  $2\,{\rm m\,s^{-1}}$  larger than at one (or more) levels anywhere higher. There is no minimum threshold for  $u_{\rm LLJ}$ . Only the lowest LLJ is counted, secondary LLJs are ignored. Due to the vertical extent of the observations up to  $290\,{\rm m}$ , only shallow LLJs are identified. Deeper or elevated LLJs, where the weak wind layer would be above  $290\,{\rm m}$ , are hence neglected.

## 3.4 Creation of an average wind profile

Even though the observations cover more than one year, the plain mean value of  $u_v(z)$  might not represent a  $10\,\mathrm{a}$  average value, e.g., some wind speeds or seasons might be over- or under-represented. Therefore the average  $10\,\mathrm{a}$  (2013-2022) wind speed (profile)  $\overline{u(z)}^{10\,\mathrm{a}}$  is calculated. A simple measure-correlate-predict model is used, where  $u_v(z)$  is correlated with  $u_v(10\,\mathrm{m})$  over the observation period (obs), which is the set of times when both values were available simultaneously; however, this is done separately for each WR and height, and the final value is a weighted average according to the frequency of the WRs. The underlying assumption is that there is a linear correlation between the wind speed at a certain height and the  $10\,\mathrm{m}$  wind speed,

140 with this correlation being different for every height and WR. The formula used is:

$$\overline{u(z)}^{10\,\mathrm{a}} = \sum_{\{\mathrm{WRs}\}} \overline{u_v(z,\mathrm{WR})}^{\mathrm{obs}(z)} \cdot \frac{\overline{u_v(10\,\mathrm{m},\mathrm{WR})}^{10\,\mathrm{a}}}{\overline{u_v(10\,\mathrm{m},\mathrm{WR})}^{\mathrm{obs}(z)}} \cdot \frac{\overline{u(10\,\mathrm{m},\mathrm{WR})}^{10\,\mathrm{a}}}{\overline{u_v(10\,\mathrm{m},\mathrm{WR})}^{10\,\mathrm{a}}} \cdot f(\mathrm{WR})^{10\,\mathrm{a}}$$
(5)

where  $\overline{u_v(z, WR)}^{\text{obs}(z)}$  is the average of  $u_v$  over obs for this height z and WR. The factor  $\frac{\overline{u_v(10\text{m},WR)}^{10\text{ a}}}{\overline{u_v(10\text{m},WR)}^{\text{obs}(z)}}$  corrects for different wind speeds during obs and 10 a. Note that  $\overline{u_v(10\text{m},WR)}^{\text{obs}(z)}$  can be different for each z. The factor  $\frac{\overline{u(10\text{m},WR)}^{10\text{ a}}}{\overline{u_v(10\text{m},WR)}^{10\text{ a}}}$  corrects for the small difference between u and  $u_v$ . The WR frequency f (Table 1) weights the corrected values as they are added. Note also that  $\overline{u_v(10\text{m},WR)}^{10\text{ a}}$  can be reduced in the nominator and denominator and that  $\overline{u(10\text{m})}^{10\text{ a}}$  is unchanged by the formula.

The WPD can be calculated accordingly. With the reduced fraction:

$$\overline{\text{WPD}(z)}^{10\,\text{a}} = \sum_{\{\text{WRs}\}} \overline{\text{WPD}_v(z, \text{WR})}^{\text{obs}(z)} \cdot \frac{\overline{\text{WPD}(10\,\text{m}, \text{WR})}^{10\,\text{a}}}{\overline{\text{WPD}_v(10\,\text{m}, \text{WR})}^{\text{obs}(z)}} \cdot f(\text{WR})^{10\,\text{a}}$$

$$(6)$$

where WPD and WPD<sub>v</sub> are the wind power density calculated from Eq. 1 using u and  $u_v$ , respectively, and  $\rho(z)$  were calculated from pressure p and temperature T at AD assuming an isothermal atmosphere and ideal gas.

One value for  $\overline{u}^{10\,\mathrm{a}}$  may lead to many different  $\overline{\mathrm{WPD}}^{10\,\mathrm{a}}$  values depending on the distribution of u over time. A broader, more variable u distribution will lead to a higher  $\overline{\mathrm{WPD}}^{10\,\mathrm{a}}$  because of the  $u^3$  dependency. To quantify this variability, a dimensionless ratio  $\eta$  between  $\overline{u}^3$ , which is proportional to WPD, and  $\overline{u}^3$  is calculated:

$$\eta(z) = \frac{\overline{u(z)^3}}{\overline{u(z)^3}} \tag{7}$$

A larger  $\eta$  means a larger variability in u. At AD, the distribution of u(z) is only available at 10 m height. At larger heights,  $\eta$  can be calculated from  $\overline{u}^{10 \text{ a}}$  and  $\overline{\text{WPD}}^{10 \text{ a}}$  instead:

$$\eta(z) = \frac{\frac{1}{2}\overline{\rho(z)} \cdot \overline{u(z)^3}}{\frac{1}{2}\overline{\rho(z)} \cdot \overline{u(z)^3}} \approx \frac{\frac{1}{2}\overline{\rho(z)}u(z)^3}{\frac{1}{2}\overline{\rho(z)} \cdot \overline{u(z)}^3} = \frac{\overline{WPD(z)}}{\frac{1}{2}\overline{\rho(z)} \cdot \overline{u(z)}^3}$$
(8)

## 3.5 Wind speed extrapolation and numerical model data


The wind speed at higher elevations can also be extrapolated from the  $10\,\mathrm{m}$  wind speed. For this the power law (Eq. 2) with  $\alpha = 0.10$  as well as the log law (Eq. 3) with  $z_0 = 0.004\,\mathrm{m}$  are used. These extrapolated wind speeds can then be used to calculate corresponding WPD and  $\eta$ .

In addition to the observations at AD (78.20196°N, 15.83369°E; 7 m altitude), corresponding data from numerical model products were analysed:

- CARRA (Copernicus Arctic Regional Reanalysis; ECMWF, 2025) provides wind speed at 10, 15, 30, 50, 75, 100, 150, 200, 250, 300, 400, 500 m height above ground with a spatial resolution of 2.5 km. In the three-hourly reanalysis the gridpoint closest to AD (index [y, x] = [515, 179] in the eastern domain) was chosen for analysis. This point is located at

the bottom of Adventdalen in the model topography even though it has a  $69 \,\mathrm{m}$  altitude there. WPD was calculated from the model wind speeds and observed T, p with Eq. 1;  $\eta$  with Eq. 7.

- NORA3 (3-km Norwegian Reanalysis; Haakenstad et al., 2021) is a hindcast with wind speed available at 10, 20, 50, 100, 250, 500, 750 m above ground every hour (lead times 4 h to 9 h) with a 3 km resolution. As for CARRA, the closest gridpoint was chosen (index [1062, 129]), located at the bottom of the valley in the model topography at 84 m altitude. WPD and η were calculated as for CARRA.
  - The Global Wind Atlas (GWA; Davis et al., 2023) provides average wind resource information at 0.25 km resolution. Wind speed and WPD values were manually read from the GWA (version 3.4) map at the grid cell containing AD. These values were available for the 10 a period 2008–2017, which is only partly overlapping with the other averaging periods (2013–2022). η was calculated from Eq. 8.

# 3.6 Weak-wind periods (WWPs)





Weak wind periods (WWPs), or lulls, are a challenge for energy supply systems which rely on wind power, especially if they are persistent. Here, a WWP is defined as  $u(10\,\mathrm{m})$  below a threshold of  $4\,\mathrm{m\,s^{-1}}$ , based on hourly values. This corresponds approximately to the time during which no, or very little, wind power can be produced. Long WWPs are especially challenging, defined as a WWP lasting at least  $4\,\mathrm{d}$ .

WWPs are calculated at six weather stations (AD, SL, JH, PL, BR, GF) in the area of Adventdalen (Table 2, Fig. 1c), with three located in the valley (AD, SL, JH) and three on the plateaus south of Adventdalen (PL, BR, GF). Data gaps up to 2 h are ignored, i.e., one or two missing values will not cut a WWP in two. As the weather stations have different measurement heights, u is extrapolated to  $10 \,\mathrm{m}$  from Eq. 3 with  $z_0 = 0.004 \,\mathrm{m}$ . This is the median  $z_0$  at AD calculated from u at  $2 \,\mathrm{m}$  and  $10 \,\mathrm{m}$  using the same equation. This extrapolation gives a new u threshold of  $3.6 \,\mathrm{m\,s^{-1}}$  at BR and  $3.4 \,\mathrm{m\,s^{-1}}$  at JH, GF at their respective measurement height (Table 2).

**Table 2.** Weather station characteristics: abbreviation and name, used variables, altitude of the ground above mean sea level, averaging period of the variables and data coverage over the 10 a period 2013 – 2022. Since all data is used with a time resolution of 1 h, an averaging period of 10 min means that the last 10 min of wind speed measurements constitute the hourly value. See Fig. 1c for their locations

| Short (long) name     | Variables              | Altitude | $\overline{u}^{10\mathrm{a}}$ (height) | Averaging period                          | Data coverage     |
|-----------------------|------------------------|----------|----------------------------------------|-------------------------------------------|-------------------|
| AD (Adventdalen)      | $u, u_v, \theta, p, T$ | 7 m      | $5.1\mathrm{ms^{-1}}(10\mathrm{m})$    | 1 h                                       | 0.98              |
| JH (Janssonhaugen)    | u                      | 251 m    | $5.1\mathrm{ms^{-1}}(3\mathrm{m})$     | 1 h                                       | 0.87              |
| SL (Svalbard Airport) | u                      | 28 m     | $5.2\mathrm{ms^{-1}}(10\mathrm{m})$    | $10\mathrm{min},$ from 2017 $1\mathrm{h}$ | 0.96              |
| PL (Platåberget III)  | u                      | 450 m    | $5.3\mathrm{ms^{-1}}(10\mathrm{m})$    | $10\mathrm{min}$                          | 0.48 (from 2018)  |
| BR (Breinosa)         | u                      | 520 m    | $4.2\mathrm{ms^{-1}}\ (5\mathrm{m})$   | 1 h                                       | 0.97              |
| GF (Gruvefjellet)     | u                      | 464 m    | $4.0\mathrm{ms^{-1}}\ (3\mathrm{m})$   | 1 h                                       | 0.69 (until 2019) |

Besides the different measurement heights, the WWP characteristics could be influenced by data gaps. This is most notable at PL and GF with several years missing. Moreover, some stations only provide an hourly  $10 \, \mathrm{min}$  average instead of the  $1 \, \mathrm{h}$  average (PL and partly LH). With a shorter averaging period, these  $10 \, \mathrm{min}$  values have a larger variance than the corresponding  $1 \, \mathrm{h}$  values, which can influence the distribution of WWPs. Most importantly durations of WWPs based on  $10 \, \mathrm{min}$  averages are shifted towards smaller values, since long WWPs will more likely be cut into several short WWPs.

#### 4 Results and Discussion




## 4.1 Characteristics of Wind Regimes

## 4.1.1 Thermally and dynamically driven Winds

Figure 2. Monthly frequency of WRs, i.e. fraction of time with the WR (Table 1) present, over 10 a (2013–2022)

Wind regimes (WRs), defined to distinguish between situations where the valley wind is *decoupled* from or *aligned* with the large-scale wind (Sect. 3.1), occur with different frequencies throughout the year. In Fig. 2 the frequency of the WRs is shown for each month. The *decoupled* WRs have a distinct seasonal pattern, with *decoupled up* very prominent in the summer months, but hardly present during the rest of the year, which is typical for anabatic (thermally driven up-valley) winds. *Decoupled down* occurs complimentary to *decoupled up*, which indicates katabatic (thermally driven down-valley) winds. Hence, the decoupled WRs are dominated by thermally driven winds. Usually this type of variability would be expected on a diurnal scale, but due to the midnight sun in summer and dark season in winter here the occurrence of thermally driven up- and down-valley winds is largely dependent on the season and less on the time of day (Henkies et al., 2023). However, these *decoupled* WRs likely

include other wind phenomena in addition to thermally driven winds. This is discernible from the occurrence of *decoupled* up in winter, when anabatic winds are not possible, as well as *decoupled down* in summer. These occurrences can instead be explained by pressure-driven channelling, where the geostrophic wind can be roughly up- or down-valley, while the valley wind is in the opposite direction (e.g., Whiteman and Doran, 1993). This has been observed, e.g., in the Rhine valley under stable stratification with a direction difference of  $100^{\circ}$  over 160 m height (Kalthoff and Vogel, 1992).

The connection of the *decoupled* WRs with thermally driven winds is in accordance with their definition (Sect. 3.1). When the large-scale wind is weak, the conditions are favourable for the development of katabatic or anabatic winds. There is no universal threshold value, especially not for a combined wind like in Adventdalen, which is a land/sea breeze and valley wind (Henkies et al., 2023).  $5\,\mathrm{m\,s^{-1}}$  is often taken as a threshold value for sea breezes (Simpson, 1994). The geopotential gradient can also serve as a threshold, e.g., Lehner et al. (2019) used a value of  $5\times10^{-4}$ , which corresponds to geostrophic wind components of ca.  $4\,\mathrm{m\,s^{-1}}$ , to determine synoptically undisturbed conditions favourable for the development of pure valley winds in the Alps. They also note that thermally driven winds can be present under stronger synoptic winds. A value of  $5\,\mathrm{m\,s^{-1}}$  therefore seems like a reasonable choice to include calm conditions. Additionally any thermally driven winds under stronger opposite large-scale winds are also contained in the *decoupled* WRs.

The across and aligned up WRs occur independent of the season. This indicates that thermally driven winds are not frequent in these regimes. Instead, dynamic drivers must cause the valley winds. Aligned down is more frequent in winter than in summer, which could be a sign of katabatic winds. However, large-scale easterly winds, which are generally stronger and more frequent in Svalbard in winter (Hanssen-Bauer et al., 2019), can also explain this seasonal variation. Since the aligned WRs occur when moderate to strong large-scale winds and valley winds roughly align, forced channelling and downward momentum transport probably dominate in these WRs over any thermal drivers. Nevertheless, across and aligned WRs can include thermally driven winds or combinations of thermally and dynamically driven winds - but much less than the decoupled WRs, which are dominated by thermally driven winds.

## 4.1.2 Wind speed profiles and LLJs in the Wind Regimes








The WRs have distinct average wind speed profiles, shown in Fig. 3. The profiles of the *decoupled* WRs have the smallest average speeds with a clear speed maximum followed by a speed decrease with height. In contrast, the *aligned* WRs have the largest wind speeds and a higher maximum with very little decrease above the maximum, while *across* has a strictly increasing speed profile.

The shape of the speed profiles is connected to the occurrence of LLJs. LLJs mark conditions with a pronounced speed decrease with height as defined in Sect. 3.3. Table 3 shows the observed LLJ characteristics. LLJs are overall frequent, with an average of 24 %. They occur in all regimes, but with different frequency and strength, which can explain the different shapes of the average wind speed profiles in Fig. 3. The large frequency and relative strength of the LLJs in the *decoupled* WRs is responsible for the clear maximum in the corresponding average speed profiles. In the other WRs, the LLJs either occur not frequent enough or are too weak compared to the overall wind speeds in that WR to cause a pronounced average maximum.

Figure 3. Average observational wind speed profiles by WR. Only complete profiles (without gaps at any heights) are included

Table 3, LLJ average characteristics for each WR and all WRs (without any weighting), only considering full profiles

| WR                          | decoupled up | decoupled down | across | aligned up | aligned down | all   |
|-----------------------------|--------------|----------------|--------|------------|--------------|-------|
| Number of profiles          | 1977         | 4154           | 1807   | 1442       | 3130         | 12510 |
| LLJ frequency               | 0.30         | 0.37           | 0.09   | 0.20       | 0.14         | 0.24  |
| $u_{ m LLJ}~{ m [ms^{-1}]}$ | 4.5          | 4.8            | 5.4    | 5.8        | 7.7          | 5.3   |
| $z_{ m LLJ}$ [m]            | 51           | 44             | 49     | 53         | 69           | 50    |

The wind speeds, the shape of the average wind profiles and the high frequency of LLJs in the *decoupled* WRs are typical for thermally driven winds in valleys (e.g., Zardi and Whiteman, 2013), which emphasizes the role of thermally driven winds in these regimes. Of these two WRs *decoupled down* has the most frequent (37 %) and shallowest LLJs with an average maximum speed of  $4.5 \,\mathrm{m\,s^{-1}}$  at 44 m height. These characteristics match with Valkonen et al. (2020), who observed a LLJ at ca. 70 m and  $5 \,\mathrm{m\,s^{-1}}$ , Mack et al. (2025), with LLJs at 31 m to  $45 \,\mathrm{m}$  and  $4.5 \,\mathrm{m\,s^{-1}}$  to  $7 \,\mathrm{m\,s^{-1}}$ , as well as Mayer et al. (2012), who determined a LLJ below 200 m height and over  $4 \,\mathrm{m\,s^{-1}}$  strength, all at AD under synoptically calm winter conditions, i.e., the *decoupled down* WR. Mack et al. (2025) also note that the LLJ is located within the stable boundary layer, which is typical for katabatic winds and explaining the small LLJ height.

The LLJ characteristics in this WR are also similar to those observed by Vihma et al. (2011) at the southern side of Advent-dalen's adjacent fjord (in the area of SL in Fig. 1c), with 65 m and 5.7 m s<sup>-1</sup> average over 20 cases. They attributed the jet mostly to a katabatic wind from the plateau on the side of the fjord, but the strength and wind direction suggest that this could well be a valley wind from Adventdalen and therefore the same jet as measured further inside in Adventdalen at AD. This is

supported by the fine scale model results of Valkonen et al. (2020), which show that in these conditions the valley jet extends
through Adventdalen and the fjord and banks against their southern side, where the measurements of Vihma et al. (2011) were
taken.

These similar observations from two locations and the supporting numerical model results indicate that the LLJ characteristics of the wind in Adventdalen are a general characteristic of the valley in the *decoupled down* WR due to thermally driven winds - rather than a local speed-up effect at AD. However, this does not necessarily imply that the exact numbers on height, frequency or strength are the same everywhere in the valley and fjord.






As for *decoupled down*, the LLJ characteristics and average speed profile of *decoupled up* are shaped by thermally driven winds, which has already been shown in detail by Henkies et al. (2023). Compared to *decoupled down*, the LLJs in *decoupled up* are slightly less frequent and at a higher elevation (51 m vs. 44 m), despite overall similarity in the average wind profiles (Fig. 3). That the observed LLJ height is larger for anabatic than for katabatic winds is typical for valley winds, although absolute heights vary greatly between valleys (e.g., Zardi and Whiteman, 2013). However, this difference is small in Adventdalen, namely 7 m considering only LLJs (Table 3) or 20 m in the average wind speed profile (Fig. 3). This can be explained by the advection of cold, stable air from the sea, which limits the anabatic LLJ height (Henkies et al., 2023).

In contrast to the *decoupled* WRs, the *aligned* WRs have an average speed increase and then a relatively constant speed with height (Fig. 3). In addition to the different profile shapes, the *aligned* WRs also generally have higher wind speeds due to their connection to relatively strong large-scale winds. The maximum height for *aligned down* is larger than for *aligned up* (140 m vs. 100 m) and *aligned down* contains the highest wind speeds. This is probably because large-scale easterly winds are frequent and because Adventdalen lies on Spitsbergen's lee side in these conditions, which can lead to a local acceleration of the wind, e.g., associated with foehn (Shestakova et al., 2022).

The LLJs in the *aligned* WRs are less frequent than in the *decoupled* WRs (14 % to 20 % vs. 30 % to 37 %, Table 3). Some of the *aligned* LLJs could be explained by the occasional thermally driven winds in these WRs, since Henkies et al. (2023) showed that the anabatic wind in Adventdalen can also develop under supporting large-scale wind. Other LLJs may be caused by various dynamic mechanisms, such as downslope windstorms or channelling, resulting in valley/fjord jets. These LLJs tend to have higher maximum speed heights, which can explain why especially *aligned down* has the highest average maximum height (69 m). Shestakova et al. (2022) showed that low-level speed-ups occur under foehn like conditions in Spitsbergen, with a case observing a jet around 150 m to 200 m height, while Sandvik and Furevik (2002) found 100 m to 150 m for a jet at the coast of Spitsbergen.

Moreover, dynamic effects, like waves, could partly explain the varying effect of LLJs on the average speed profiles. Stationary waves, especially in the stable boundary layer, can occasionally accelerate, but also decelerate the flow at different levels. This is due to the generally variable wavelength, which depends on the complex interplay of stability, background flow and terrain (Richner and Hächler, 2012). When such acceleration or deceleration occurs over periods sufficient to show in the 1 h averages, this change will increase the LLJ frequency. However, the effect of waves on the wind speed will likely cancel out over long averages. Hence, unlike thermally driven winds, which increase both the LLJ frequency and the average low-level

wind speed in the *decoupled* WRs, the varying wave effect might lead to an increased LLJ frequency while not really affecting the average wind speed profile in the *aligned* WRs.

Across is the only WR where the wind speed increases strictly with height and where LLJs occur most rarely (9%). This is a sign that the wind is rarely channelled and speed up at low-levels when it is crossing the valley. Any channelling that occurs is caused by the much smaller profile of a sidevalley and hence more unlikely than channelling along the broad main valley Adventdalen. In addition, thermally driven winds are very unlikely when the wind is not following the main valley axis.

#### 4.2 Long term general characteristics


## 290 4.2.1 Average wind speed profile and LLJs

All observed wind speed profiles were combined into a  $10\,\mathrm{a}$  observational wind speed profile (Sect. 3.4). Figure 4 shows this observational profile along with the wind speed profiles of the power law (Eq. 2) and the log law extrapolation (Eq. 3) and of the numerical models CARRA, NORA3 and GWA (Sect. 3.5). The observational profile has a speed maximum at  $80\,\mathrm{m}$  to  $100\,\mathrm{m}$  and above a speed decrease by  $5\,\%$  towards  $290\,\mathrm{m}$ , while all other wind speed profiles have a similar shape, increasing monotonically with height.

Figure 4. 10 a average wind speed u profiles based on observations, the power law with  $\alpha = 0.10$ , the log law with  $z_0 = 0.004$  m and the numerical models CARRA, NORA3 and GWA. All 10 a averages cover the period 2013-2022, except GWA with 2008-2017 (Sect. 3.5)

The shape of the observational profile is explained by the large frequency of the *decoupled* WRs, which occur 14% + 32% = 46% of the time (Fig. 2). This shows the overall relevance of thermally driven winds for the wind speed profile in Adventdalen.

However, this 10 a maximum is higher than in the *decoupled* WRs due to the influence of the other WRs without a distinct speed maximum. Especially the *aligned* WRs increase the overall wind speed.

The average 10 a wind profile in Adventdalen with its distinct speed decrease above 100 m is different from those found in many other locations. Moreover, the occurrence of LLJs with an average of 24 % (Table 3) is higher than elsewhere. For example, Lampert et al. (2016) show that at a location in the Northern German Plain the 1 a average wind speed was strictly increasing to 420 m and almost constant above up to 500 m. At the same time LLJs were found between 7 % and 11 % of the time. Similar LLJ frequencies were found at the German North Sea coast by Rausch et al. (2022). Long-term measurements from various places in Germany, Denmark and the North sea also reveal strictly increasing average wind speeds with height (e.g., Gryning et al., 2016).

Moreover, the average LLJ height of  $50 \,\mathrm{m}$  in Adventdalen is much lower than in northern Germany with  $110 \,\mathrm{m}$  to  $120 \,\mathrm{m}$  at the coast and  $200 \,\mathrm{m}$  to  $220 \,\mathrm{m}$  inland. The average speed of  $5.3 \,\mathrm{m\,s^{-1}}$  in Adventdalen is not directly comparable, but well within the ranges of  $4 \,\mathrm{m\,s^{-1}}$  to  $9 \,\mathrm{m\,s^{-1}}$  (inland) or  $2.5 \,\mathrm{m\,s^{-1}}$  to  $17 \,\mathrm{m\,s^{-1}}$  (coast).

The explanation for the difference in LLJ height and frequency can be found in the driving mechanisms. Section 4.1.2 has shown that the LLJs in Adventdalen are mostly thermally driven and located within the stable boundary layer and sometimes driven by dynamic topographic effects. In contrast, LLJs over plains are often explained by inertial oscillations, where the LLJs are located atop an inversion (e.g. Banta et al., 1993). Sometimes inertial oscillations are even used as the premise for LLJ analysis in northern Germany (e.g., Emeis, 2014b). Despite different overall characteristics, the LLJs in Adventdalen and northern Germany have in common that the LLJ occurrence heavily depends on the large-scale weather (Emeis, 2014b). Although the LLJs in Adventdalen are thermally driven to a large degree, it is at least plausible that inertial oscillations could play a role as well. This is because mixed thermally and inertially driven forms of LLJs can exist in moderately complex terrain, such as occasionally in very broad valleys (e.g., Chiao and Dumais, 2013) or in coastal areas where the coast-parallel LLJs is a result of the thermal wind balance between the local temperature gradient and the Coriolis force (e.g., Parish, 2000).

## 4.2.2 Extrapolation laws and numerical model data







The unusual shape of the observational wind speed profile (Fig. 4) makes it difficult to extrapolate from the  $10\,\mathrm{m}$  wind speed using simple laws. Both the power law (Eq. 2) with  $\alpha=0.10$  and the log law (Eq. 3) with  $z_0=0.004\,\mathrm{m}$  create profiles very similar to each other and strictly increasing speed with height (Fig. 4). Both overestimate the observed wind speed at all heights and especially at larger heights. This is understandable as these laws were made for the well mixed constant-flux layer in the lowest 10s of metres and not for these heights. Adjusting  $\alpha$  and  $z_0$  would improve the result, at least up to the wind speed maximum. For the power law  $\alpha=0.07$  would be necessary even though this is untypically small. Choosing a much smaller  $z_0$  would physically mean an extremely smooth surface like ice, which is unrealistic at the location.

Both simple laws ignore stability effects. Stable conditions usually lead to a stronger increase of wind speed with height, as seen by a larger  $\alpha$  value in stable conditions (e.g., Sisterson et al., 1983). This would make the results from the extrapolation methods even less realistic. The explanation for this paradox is that the stable conditions in Adventdalen are correlated with thermally driven winds, which increase the wind speed at low levels (Sect. 4.1.2).

Further, the average wind speeds of CARRA, NORA3 and the Global Wind Atlas (GWA) are shown in Fig. 4. All three profiles have a strong and monotonous wind speed increase with height, GWA most and NORA3 least. They underestimate the observed wind speeds at levels below 100 m and overestimate them above 100 m, GWA the most. At 100 m they are within  $0.3\,\mathrm{m\,s^{-1}}$  of the observations, since this is approximately where the profiles cross each other. The reason for the profile shapes of CARRA and NORA3 is probably their resolution of 2.5 to 3 km, which does not resolve the flat valley floor and steep walls of Adventdalen (Sect. 3). Hence, the models see the topography of Adventdalen smoothed out, but with a rougher surface. Valkonen et al. (2020) and Mack et al. (2025) stress the effect of the model resolution showing that the 0.5 km model performed better in representing the LLJs in Adventdalen than the operational 2.5 km model. In addition, models can in some cases underestimate the strength of a LLJ at low levels (e.g., Storm et al., 2009).

GWA has a much finer resolution (0.25 km) than CARRA and NORA3, but it still cannot reproduce the shape of the observations. This is because the GWA with its finest resolution is not a complete weather model, which would be able to represent local, thermally driven winds. Instead, it only accounts for some dynamic effects of the terrain like local speed-ups, while the general wind characteristics are still based on a mesoscale model with a much larger resolution on the kilometre scale (Davis et al., 2023).

#### 4.3 Wind power density and wind variability




Figure 5. 10 a average WPD profiles, otherwise the same as Fig. 4

Wind power density (WPD) profiles are shown in Fig. 5 for the observations, extrapolations and numerical models. The observed WPD profile has a broad maximum at  $120 \,\mathrm{m}$  to  $140 \,\mathrm{m}$ , above which WPD decreases with height by up to  $5 \,\%$  towards  $290 \,\mathrm{m}$ . This is similar to the shape of the wind speed profile, as seen in Fig. 4, except that the WPD maximum is

slightly higher and broader. The WPD values for all other profiles than the observations in Fig. 5 are, like the corresponding wind speed profiles in Fig. 4, strictly increasing. The WPDs at heights of 100 m and above are all larger than the observations, while the values are more similar towards the smallest heights. The only exceptions are CARRA, which has smaller values than the observations at low heights, and GWA, which has larger values everywhere. Overall, these WPD profiles overestimate the observational WPD more than they do for the wind speed.




Both the observed wind speed and WPD decrease by 5% from maximum to 290 m height. However, due to the dependence of the WPD on the cube of the wind speed (Eq. 1), the WPD should be expected to be reduced by more, i.e.,  $1-(1-5\%)^3=14\%$ . That the reduction is only 5%, can be explained by the variability  $\eta$  (Eq. 7), which connects the wind speed and WPD. The profiles of  $\eta$  are plotted in Fig. 6 for the observations, extrapolations and numerical models. In the observations,  $\eta$  increases with height, so that at 290 m there is a larger share of relatively high (and also relatively low) wind speeds than at, e.g., 100 m. The larger share of high wind speeds makes the WPD at 290 m larger than one would expect from the change of the average wind speed.

Figure 6. Variability  $\eta$  profiles, otherwise the same as Fig. 4. Note that the power and log law have identical values to the observations at 10 m because they are extrapolated from this value

The thermally driven winds in Adventdalen have the strongest influence at low levels, where they significantly increase wind speed, particularly during otherwise calm conditions. This suggests that these winds play a key role in reducing wind variability at low levels. Without the thermally driven valley winds, wind variability would likely decrease with height. This pattern has been observed in other regions, such as the Northern German plains, where wind speed variability decreases above ca. 140 m (Lampert et al., 2016).

 $\eta$  is higher for the numerical models than for the observations (Fig. 6). While CARRA and NORA3 have increasing  $\eta$  with height, like the observations, GWA has the highest  $\eta$  values close to the ground. This shows the limited ability of the GWA to accurately represent the wind climate in Adventdalen with its frequent thermally driven winds.

#### 370 4.4 Weak-wind periods (WWPs)

**Figure 7.** Characteristics of WWPs at individual weather stations (see locations in Fig. 1) and at the combination of AD and PL: (a) average occurrence frequency of WWPs; (b) normalized distribution (histogram) of occurrence frequency by WWP duration, using bins of 6 h; (c) mean and standard deviation of the WWP yearly maximum durations; (d) average occurrence frequency of WRs during long WWPs (at least 4 d)

The previous section has shown that the thermally driven winds can make the wind in Adventdalen less variable on average at low levels. This section examines if this also affects the length of the periods with little wind. Weak wind periods (WWPs)

with winds below  $4\,\mathrm{m\,s^{-1}}$ , when very little wind power can be produced, are characterised by frequency and duration, which are displayed in Fig. 7 for six individual weather stations and one weather station combination. The overall WWP frequency is shown in Fig. 7a. Moreover, the frequency of WWPs with a certain length can be seen (Fig. 7b) at the weather stations AD and PL, which represent valley and plateau, respectively. The durations of yearly extreme WWPs are shown in Fig. 7c with mean and standard deviation. Figure 7d visualizes the frequency of long WWPs (at least  $4\,\mathrm{d}$  long) and which WRs occur during these long WWPs.

WWPs are present 35% to 53% of the time. These values are similar to those Leahy and McKeogh (2013) observed at several locations in Ireland when considering the  $80 \,\mathrm{m}$  wind speed and a threshold of  $6 \,\mathrm{m\,s^{-1}}$ . The valley weather stations (AD, JH, SL) have a slightly lower fraction of WWP than the plateau stations (PL, BR, GF).

However, there are important differences in the duration of the WWPs between valley and plateau (Fig. 7b): While AD (valley) has a higher share of short WWPs, long durations of several days are more common at PL (plateau). The difference between valley and plateau becomes even clearer when looking at the yearly maximum duration of WWPs (Fig. 7c): the weather stations on the plateaus have larger mean values (5 d to 8 d) than those in the valley (ca. 3 d). Moreover, long WWPs are more frequent on the plateaus (Fig. 7d).

The WWPs occur mostly during the *decoupled* WRs, indicating weak large-scale forcing and potential conditions for thermally driven winds. This means that during these events the valley wind is sometimes strong enough to disrupt the WWP, while the wind on the plateaus is not. This does, however, not mean that the wind speed in Adventdalen is always above threshold when there are WWPs at the plateaus, but rather that the local valley winds overcome the threshold occasionally, cutting a long WWP into several shorter ones.

Combining two weather stations, the occurrence and length of WWPs is reduced, i.e. sometimes a WWP occurs only at one station while not at the second, and sometimes it is the other way around. This is shown for the combination of AD and PL in Fig. 7. There are situations in which the valley wind is stronger than the wind at the plateau, but also the opposite is possible. These situations, with calm conditions in the valley and windy at the plateau, are typically driven by the large-scale wind, when the valley is just more sheltered from the wind. This is not only true for the combination AD+PL, but for any weather station combination (not shown). This highlights that geographical diversity reduces the combined WWPs, which has been confirmed in other places by observations (e.g., Leahy and McKeogh, 2013) and models over whole countries such as the UK (e.g., Cannon et al., 2015).

#### 400 5 Implications for wind energy and conclusions





It has been shown that several characteristics of the wind in Adventdalen can be considered beneficial for wind energy in High Arctic valleys. First, LLJs are frequent in Adventdalen due to local, thermally driven valley winds (Sect. 4.1.2). While the strong shear associated with usual LLJs is a challenge for wind turbines, the negative shear above shallow LLJs can have positive effects on turbine loads (Gutierrez et al., 2017). Nevertheless, the stable conditions associated with the thermally driven

LLJs can also reduce the extractable wind power in a wind farm slightly more than usual due to reduced wake recovery (e.g., Barthelmie and Jensen, 2010; Emeis, 2010).

Second, LLJs shape the average wind profile to the extent that there is a wind speed maximum around 80 m to 100 m height (Fig. 4) and a broad WPD maximum around 120 m to 140 m (Fig. 5). Therefore, there is no clear advantage of building very high wind turbines, e.g., at 200 m hub height instead of 100 m as the WPD is almost the same at both heights. On the contrary, the wind at 100 m is less variable than at higher altitudes, where both very low and high wind speeds are more frequent (Fig. 6), allowing a more even extraction of wind power at 100 m. While the exact heights may differ for different valleys and locations therein, the same pattern can be expected in other large Arctic valleys with frequent decoupling. Using small or medium-sized turbines can also make transport and construction easier, especially in the Arctic due to the widespread permafrost and remoteness.




Considering the placement of small turbines at 10 m height, Adventdalen has some advantages over the plateaus, since the reduced wind speed variability shortened the WWPs in the valley compared to plateau locations, especially those of greatest durations. Besides, valleys like Adventdalen generally have more favourable icing conditions and road access compared to mountain sites, which a feasibility study also highlighted (Ludescher-Huber et al., 2023). However, a combination of sites with diverse wind characteristics can potentiate the benefits, e.g., turbines on plateau or mountain level can harvest considerable wind energy in high-wind conditions, while turbines in Adventdalen supply the needed power when the elevated locations are affected by WWPs or icing. This would create redundancy and decrease the necessary energy storage size, increasing energy security in offgrid settings such as Svalbard with potentially high wind power penetration.

The comparison of long-term numerical model data with the observations has revealed that the models do not accurately represent the wind profile and overestimate the wind resource in Adventdalen at heights of  $100\,\mathrm{m}$  and more, while they overestimate the wind variability at lower heights. This highlights the need for both models with a suitable horizontal resolution in the hectometre scale and observational data at potential sites as well as a good understanding of the physical processes in complex Arctic terrain.

To conclude, the observations from Adventdalen have shown that thermally driven winds play an important role in the wind characteristics there. This is because they create LLJs frequently, increase the average wind speed at low levels and reduce the length of periods with little wind.

Data availability. The Sodar and Lidar data can be found under Henkies et al. (2024a, b). The weather station data was downloaded from The University Centre in Svalbard (https://www.unis.no/facilities/weather-stations/; AD, JH, BR, GF) and from the Norwegian Meteorological Institute via the frost API (https://frost.met.no/index.html; PL, SL). ERA5 is available at Hersbach et al. (2018). The CARRA data can be found under Schyberg et al. (2021a, b). NORA3 data was downloaded from https://thredds.met.no/thredds/projects/nora3.html. The Global Wind Atlas can be found at https://globalwindatlas.info/en/.

*Author contributions*. M. Henkies did the conception of the study, the data analysis and the figure and manuscript preparation. A. Sjöblom, K. V. Høyland and A. Shestov provided detailed feedback on the concept, analysis, figures and manuscript.

Competing interests. The authors declare that they have no conflict of interest.

- Acknowledgements. The authors want to thank everyone involved in the collection of wind profile data that was used in this study, especially

  Stephan Kral and Lukas Frank for providing expert knowledge on the Sodar and Lidar data, as well as Lara Ferrighi at the Arctic Data Centre
  for help with publishing the data and Steve Coulson for comments on the manuscript.
  - This work was partly funded by ZEESA Zero Emission Energy Systems for the Arctic (Research Council of Norway / RCN, project number 336342). The data collection was funded by the National Norwegian Research Infrastructure OBLO Offshore Boundary Layer Observatory (RCN, 227777), OBLEX Offshore Boundary-Layer Experiment (RCN, 807770) and Data collection for validating the AROME-Arctic
- 445 Weather Model. RiS ID: 11430 (RCN, 310733).

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
