# Peer review of "The Influence of Local Winds on Wind Power Characteristics in a High Arctic Valley"

_Wind Energy Science, 2025_

## Author Comment (AC1)

**Author responses to the reviewer comments**

**of the manuscript "The Influence of Local Winds on Wind Power Characteristics in a High Arctic Valley" (wes-2025-61) by Matthias Henkies, Knut Vilhelm Høyland, Aleksey Shestov, and Anna Sjöblom**

We thank the referees for taking the time und putting in the effort to carefully read our manuscript and give detailed and constructive feedback on it.

We have tried to address all of the raised points in the following. The comments, which are numbered consecutively, are written in *italics*, followed by our answer and our changes in the manuscript, with added words underlined and in blue and deleted words crossed out and in red.

**Referee 1 (RC1)**

**General comments:**

*It is an interesting and careful analysis of the complex flow in the Advent valley – a flow that has been the focus of several publications throughout the years.*

*The manuscript is focused on wind energy and the conclusion is interesting – that the optimum for wind energy production is wind turbines located in the valley with a hub height of approximately 100 m, thus taking advantage of the frequent LLJ in the valley.*

*I have a few points:*

**Comment 1:**

*(1): As written in the text it could be understood as if A is the sweeping area of the wind turbine and thus covering heights with varying wind speed. The WPD is the available power for a unit area at a given height. Please clarify this in the text.*

**Answer 1:**

We agree with the reviewer that our formulation is not clear enough, so we have tried to make it more precise.

**Changes 1:**

"Wind power density (WPD) is the kinetic power *P* of the wind crossing a unit area *A* at a given height *z*. WPD is very sensitive to the wind speed *u*, which varies with  *z*" (l. 66f)

**Comment 2:**

*(3): The equation for the log wind profile is: u(z) = (ustar/kappa) * ln(z/zo) thus kappa in Eq.(3) is misplaced (inverse). It is not clear if this is just a typo or it will have implications on the use of the log law for the rest of the paper.*

**Answer 2:**

We thank the reviewer for noticing this typo, which did not affect any calculations. It has now been changed.

**Changes 2:**

$$\frac{u(z)}{\kappa u_*} \frac{\kappa u(z)}{u_*} = \log\left(\frac{z}{z_0}\right)$$

(Eq. 3; l. 84)

**Comment 3:**

*Line 114: It is not clear what is meant by "at least 9 min of measurements with a 10% availability". This suggests, although it is stated that the analysis is based on 10 min averages, that the analysis is based on the original raw 1 Hertz data from the wind lidar? What is the threshold CNR? 10% availability is very low – I suggest repeating the analysis with an availability of say 50% and see if this changes the result and conclusions. A lower CNR threshold combined with higher data availability might be preferable.*

**Answer 3:**

The authors agree that this can be seen as unclear and that different threshold settings can be advantageous in some situations.

The analysis is indeed purely based on 10 min averages output by the Lidar instrument itself. These 10 min files contain a flag stating whether the instrument was actually running for at least 9 min. This is to exclude extremely rare cases where, e.g., the Lidar was restarting after a power cut; in practice only 184 Lidar values (0.14 %) were excluded due to this flag.

The CNR threshold was set to -22 dB and it is unfortunately impossible for us to use a different CNR threshold. Either way it is a good idea to check whether a different availability threshold of 50 % instead of 10 % would change the results. The number of available Sodar and Lidar 1 h values would be reduced by 1.21 %. The effect on the observed LLJ quantities is seen in the

following table, which is a modification of Table 3 in the original manuscript. It shows both the original 10 % (black) and a 50 % availability threshold (blue underlined):

| WR | decoupled up | decoupled down | across | aligned up | aligned down | all |
|---|---|---|---|---|---|---|
| Number of profiles | 1977 (1961) | 4154 (4072) | 1807 (1783) | 1442 (1382) | 3130 (3066) | 12510 (12264) |
| LLJ frequency | 0.30 (0.30) | 0.37 (0.36) | 0.09 (0.09) | 0.20 (0.20) | 0.14 (0.14) | 0.24 (0.24) |
| $u_{LLJ}$ [m s$^{-1}$] | 4.5 (4.5) | 4.8 (4.8) | 5.4 (5.4) | 5.8 (5.8) | 7.7 (7.5) | 5.3 (5.3) |
| $z_{LLJ}$ [m] | 51 (52) | 44 (44) | 49 (49) | 53 (53) | 69 (69) | 50 (50) |

The table shows that the LLJ characteristics would differ only slightly, hence the results regarding LLJs would not change under the 50 % threshold.

The effect of the threshold change on the observed profiles of $u_v$, $u$, WPD and $\eta$ is seen in the altered figures on the next page. The wind speed profiles of some wind regimes would be slightly modified at high levels, while their overall shape would be unchanged (Alt. Fig. 3). The long-term average profiles of $u$, WPD and $\eta$ based on 50 % availability would be almost identical to the original ones based on 10 % (Alt. Fig. 4, 5, 6) due to the compensating effect of the formula used for the creation of an average wind / WPD profile. This means that also these results as well as the manuscript's conclusions would be unchanged.

To conclude, while the exact numbers might be different at the last digits depending on the choice of availability threshold, the choice does not influence the shape of the observed profiles or the LLJ characteristics. Since it is not possible in this case to simultaneously lower the CNR threshold and increase the availability threshold, we choose to leave the availability threshold at 10 %. To avoid confusion, we added the CNR threshold in the manuscript and tried to explain the filtering due to power cuts more thoroughly.

**Changes 3:**

"The Lidar was able to measure during high wind speed cases, while, with a CNR threshold of -22 dB, it occasionally had problems in bad visibility, independent of the wind speed." (l. 111f)

"Only data Intervals during which the Lidar was only partly running, e.g., due to a power cut, were excluded based on a flag in the 10 min data files. In addition only measurements with at least 9 min of measurements and 10 % data availability were used." (l. 114 – 116)

(None of the tables or figures in the manuscript were changed.)

[Figure]

Altered Figure 3

[Figure]

Altered part of Figure 4

[Figure]

Altered part of Figure 5

[Figure]

Altered part of Figure 6

**Comment 4:**

*I note that motions caused by waves in the stable atmosphere of the valley also might contribute to the flow pattern. Please discuss.*

**Answer 4:**

We agree that the effect of waves is worth discussing as they are common in the stable atmosphere. With an averaging period of 1 h, we might see the effect of stationary waves on the wind speed profile. Therefore we included a paragraph in the manuscript which discusses the effect stationary waves might have on the LLJ frequency and the average wind speed profile.

**Changes 4:**

"Moreover, dynamic effects, like waves, could partly explain the varying effect of LLJs on the average speed profiles. Stationary waves, especially in the stable boundary layer, can occasionally accelerate, but also decelerate the flow at different levels. This is due to the generally variable wavelength, which depends on the complex interplay of stability, background flow and terrain (Richner & Hächler, 2012). When such acceleration or deceleration occurs over periods sufficient to show in the 1 h averages, this change will increase the LLJ frequency. However, the effect of waves on the wind speed will likely cancel out over long averages. Hence, unlike thermally driven winds, which increase both the LLJ frequency and the average low-level wind speed in the *decoupled* WRs, the varying wave effect might lead to an increased LLJ frequency while not really affecting the average wind speed profile in the *aligned* WRs." (l. 278 – 285)

"Richner, H. and Hächler, P.: Understanding and Forecasting Alpine Foehn, in: Mountain weather research and forecasting: recent progress and current challenges, Springer, New York, ISBN 978-94-007-4097-6, 2012." (in the references)

**Referee 2 (RC2)**

**General comments**

*This study investigates the influence of local winds on wind power characteristics in a Svalbard valley, using long-term observations from a wind profiler, lidar, and weather stations. The observations show the development of various wind systems, driven by thermal gradients, the terrain and synoptic conditions. Different wind regimes are discussed in detail. They conclude that a wind speed maximum was observed at around 80–100 m, and the wind power density maximum at 120–140 m. This is useful information for setting up wind power stations in an Arctic valley and the authors provide valuable recommendations for different locations. The observations analyzed in this study are of high value, as the boundary layer structure and its wind field in such complex, non-homogeneous terrain are still not well understood and adequately parameterized in models. The study supports the scientific community in deepening the understanding of the polar boundary layer structure and wind fields in complex terrain. This is particularly important given the lack of knowledge due to the lack of boundary layer data from Arctic mountainous regions and the limitations of numerical models, which often perform inadequately in such environments and conditions, which is also shown in the paper*

*The introduction is clear and provides a good overview of previous boundary layer studies in this area. The theoretical background is well described, and the methodology is sound. The Adventdalen valley is a suitable experimental site, with long-term wind data at different sites that allow for seasonal analysis and the various observing locations give a good spatial coverage. Key data have already been published by the authors, and other datasets are openly available. To summarize: the research question is both important and original. The conclusions regarding the placement of wind power stations and the types of turbines suitable for a high Arctic valley are well supported by the data. The paper effectively demonstrates the importance of using local wind observations in complex terrain, rather than relying solely on models or wind atlas data, which often have a too coarse resolution.*

*The paper is well suited for publication in Wind Energy Science Discussions. The topic is timely and relevant, and the methodology is robust. The text is generally well written.*

*I do not have any major corrections/suggestions, just minor*

*technical corrections:*

**Comment 5:**

*Line 89, page 5: The word "still" sounds a bid odd at the beginning of the sentence. I suggest replacing it with something like: "It has to be kept in mind that…".*

**Answer 5:**

We agree and adapt the sentence.

**Changes 5:**

" It has to be kept in mind that these two laws are based on measurements in flat terrain and may not be applicable in a valley like Adventdalen." (l. 89)

**Comment 6:**

*Line 84: the von Karman constant is written inverted incorrectly; it should be U(z)/U\*=1/k log(z/zo).*

**Answer / changes 6:**

We thank the reviewer for noticing this typo. It has now been changed (see comment 2).

**Comment 7:**

*Lines 199, 204: Please include definitions of katabatic and anabatic winds for clarity.*

**Answer 8:**

The paragraph now includes the definition of the anabatic and katabatic valley winds.

**Changes 9:**

"... which is typical for anabatic (thermally driven up-valley) winds." (l. 200)

"... which indicates katabatic (thermally driven down-valley) winds." (l. 201)

**Comment 8:**

*Figure 4, 5,6: Did you have in your calculation code for the logarithmic power law, which is shown in the graphs, the von Karman constant correct or inverted (see previous point line 84)?*

**Answer / changes 8:**

The typo in the logarithmic law did not affect any calculations; see answer 2.

**Comment 9:**

*Figure 7 caption: I suggest including in the caption of Figure 7 after "individual weather stations," "see locations in Figure 1".*

**Answer 9:**

We adapt this suggestion in the caption of Fig. 7.

**Changes 9:**

"**Figure 7.** Characteristics of WWPs at individual weather stations (see locations in Fig. 1) and at the combination of AD and PL: ..." (caption of Fig. 7)